# Development and Characterization of Glass-Ceramics from Combinations of Slag, Fly Ash, and Glass Cullet without Adding Nucleating Agents

**DOI:** 10.3390/ma12122032

**Published:** 2019-06-25

**Authors:** Diana M. Ayala Valderrama, Jairo A. Gómez Cuaspud, Judith A. Roether, Aldo R. Boccaccini

**Affiliations:** 1Grupo de Física de Materiales, Universidad Pedagógica y Tecnológica de Colombia. Av Central del Norte 39-115, 150003 Tunja, Boyacá, Colombia; jairo.gomez01@uptc.edu.co; 2Institute of Biomaterials, Department of Materials Science and Engineering, University of Erlangen-Nuremberg, 91058 Erlangen, Germany; aldo.boccaccini@ww.uni-erlangen.de; 3Institute of Polymer Materials, Department of Materials Science and Engineering, University of Erlangen-Nuremberg, 91058 Erlangen, Germany; judith.roether@ww.uni-erlangen.de

**Keywords:** glass-ceramics, fly ash, glass cullet, slag

## Abstract

Developments in the field of materials science are contributing to providing solutions for the recycling of industrial residues to develop new materials. Such approaches generate new products and provide optimal alternatives to the final disposal of different types of industrial wastes. This research focused on identifying and characterizing slag, fly ash, and glass cullet from the Boyacá region in Colombia as raw materials for producing glass-ceramics, with the innovative aspect of the use of these three residues without the addition of nucleating agents to produce the glass-ceramics. To characterize the starting materials, X-ray diffraction (XRD), X-ray fluorescence (XRF), and Scanning Electron Microscopy (SEM) techniques were used. The results were used to evaluate the best conditions to produce mixtures of the three waste components and to determine the specific compositions of glass-ceramics to achieve products with attractive technical properties for potential industrial applications. The proposed mixtures were based on three compositions: Mixture 1, 2, and 3. The materials were obtained through thermal treatment at 1200 °C in a tubular furnace in accordance with the results of a comprehensive characterization using thermal analysis. The microstructure, thermal stability, and structural characteristics of the samples were examined through SEM, differential thermal analysis (DTA), and XRD analyses, which showed that the main crystalline phases were diopside and anorthite, with a small amount of enstatite and gehlenite. The obtained glass-ceramics showed properties of technical significance for structural applications.

## 1. Introduction

The incorporation of industrial waste in production processes is receiving increasing interest, given that some industrial waste materials have physical–chemical properties that can be exploited to obtain new materials for technological or industrial applications, providing a better alternative to environmentally unfavorable waste disposal. In the specific case of current work, the region of Boyacá (Colombia) has different thermoelectric power plants, steel industries, and many glass industries that are responsible for the generation of 800,000 tons of fly ash per year, in addition to other industrial wastes such as slag and glass cullet, of which at least 50,000 tons correspond to the Boyacá region [1]. For each ton of manufactured steel, a total of 0.1–0.3 tons of slag could be generated, reaching an average of 70,000 to 96,000 tons per year [2]. The elemental compositions of the fly ash vary according to the type of coal, the degree of pulverization, and the type of collectors used, even when the fly ash is obtained from the same source [3]. Previous studies have shown that the fly ash generated by Boyacá industry is classified as F class, in accordance with X-ray fluorescence analysis and ASTM C 618 Standard specifications [4,5,6]. On the other hand, slag forms from the fusion of mineral impurities added during steel production, and its extraction occurs at 1400 °C, generating glassy products rich in CaO, SiO_2_, MgO, MnO, and Fe_2_O_3_ [7,8]. In the case of glass cullet, approximately 5000 tons per year are derived from consumption and production processes in the Boyacá region: This residue can be completely recycled in order to exploit alternatives to its final disposal. It is evident, in accordance with the amount of waste generated in Boyacá, that these residues could be used as additives in the production of asphalt and concrete [9,10]. As a technical alternative, research has focused on the use of such residues in obtaining glass-ceramics, bearing in mind that these materials may have attractive properties for applications in several industrial sectors. Additionally, glass-ceramic technology represents a versatile (materials science) approach to immobilizing various types of radioactive and dangerous wastes. However, due to the high melting temperatures and special heat treatment conditions that are required to reach adequate properties of glass-ceramics, the use of waste can only be justified if high-value products with suitable properties for industrial applications can be developed [8,11,12].

The design of mixtures for the development of glass-ceramics needs careful control through composition and phases that will determine the physicochemical and mechanical properties of the materials. During elemental composition, oxides that act as network formers are required, specifically silicon oxide (SiO_2_) and phosphorus oxide (P_2_O_5_). Network-modifying oxides such as sodium oxide (Na_2_O), potassium oxide (K_2_O), magnesium oxide (MgO), calcium oxide (CaO), and oxides that act as intermediates (e.g., alumina (Al_2_O_3_), titanium oxide (TiO_2_), or ferric oxide (Fe_2_O_3_)) are necessary to stabilize the structure [13]. According to Erol et al. [14], a significant amount of Fe_2_O_3_ can be used as a nucleating agent in the production of glass-ceramics, providing an important advantage in developing a great variety of microstructures with different morphological conformations [14,15]. Differences in appearance are also considered: They depend on crystal growth characteristics related to thermal treatment for nucleation and crystallization, properties that can be improved for technological, industrial, or domestic applications. In particular, the mechanical properties of glass-ceramics derived from this kind of waste mixture can achieve high values [16,17]. In terms of electrical and magnetic properties, several studies have reported the relevant functional properties of glass-ceramics, including waste-derived glass-ceramics, for technological applications [16,18,19]. For these reasons, the present work focused on the synthesis and characterization of three glass-ceramics based on a combination of the mentioned wastes, aiming at developing added value glass-ceramic products for technical applications.

## 2. Results

### 2.1. Characterization of Raw Materials

The morphology of raw materials evaluated by scanning electron microscopy (SEM) confirmed the presence of heterogeneous particles in the eight fly ash samples, corresponding with aluminosilicate agglomerated particles, as shown in Figure 1. The morphology of the slag and glass cullet can be observed in Figure 2a–c.

The elemental compositions of the three raw materials of the fly ash, slag, and glass cullet were evaluated by means of X-ray fluorescence (Figure 3 and Figure 4).

### 2.2. Preparation, Analysis, and Design of Mixtures

With the selected materials (fly ash H, slag 2, and glass cullet), three mixtures were prepared according to Table 1. Figure 5 shows the schematic heat treatment and annealing processes performed to obtain glass-ceramic materials. The heating rates were chosen based on the results obtained by DTA analysis (Figures 8–10). The nominal compositions of the mixtures are shown in Table 2.

The obtained multicomponent systems were characterized using XRD analysis and Rietveld refinements, and the results are shown in Figure 6a–c. In these figures, the symbol “X” represents the experimental diffraction pattern, the red line is the calculated diffractogram, the green line corresponds with the background, and the blue line is the difference between the experimental and the calculated patterns. The theoretical diffraction patterns to perform the refinement were obtained from a crystallography comparative database. The presence of quartz and mullite was confirmed except for the Mixture 3, which exhibited an Al_2_O_3_ phase, as shown in Figure 6.

SEM images of the mixtures (shown in Figure 7) confirmed the morphology and microstructure after the grinding and mixing processes, which were characterized by particles of fly ash as spheres and were mainly associated with the presence of aluminosilicate phases derived from coal combustion processes. In addition, the presence of agglomerates was associated with the slag, and finally, high heterogeneity was observed without the defined tendency that was associated with the glass cullet component.

### 2.3. Glass-Ceramic Synthesis

#### 2.3.1. Thermal Analysis (DTA-TGA)

The glass-ceramic mixtures were subjected to thermal treatment at 1200 °C for 2 h, with a heating ramp of 5 °C min^−1^, which was selected to obtain an amorphous material as the starting point for the sintering of glass-ceramic materials. The corresponding TGA-DTA analyses are shown in Figure 8, Figure 9 and Figure 10.

#### 2.3.2. Sintering

After the first thermal treatment of the samples, milling and cold pressing processes were carried out, and pellets ~14 mm in diameter with ~2.5 mm thickness were obtained using Glycerol BioXtra ≥99% (5% by weight) for the compaction of the powders at a compressive stress of 30 MPa. Then, the pellets were sintered, taking into account the temperatures indicated in Table 3, which were determined in accordance with the DTA results for each material.

The three mixtures analyzed by XRD presented different crystallographic features (Figure 11, Figure 12 and Figure 13), indicating that the three mixtures had the same crystalline phases, but with different relative contents.

The morphology and surface appearance of the three samples corresponded to irregular agglomerates distributed heterogeneously. The samples were seen to be made up of particles of different sizes, as observed in the SEM images of each material obtained before and after the sintering treatment (Figure 14).

The mechanical strength was determined using four samples of each type at a constant displacement speed of 0.5 mm min^−1^ for each of the three mixtures and was calculated by the following relationship (Equation (1)), which corresponds to the diametral compression (“Brazilian”) test [20]:
σ_T_ = 2*P*/(π·*d*·*h*),(1)
where *P* is the load at failure (N), *d* is the diameter of the test pellet (mm), and *h* is the thickness of the test pellet (mm). The values of the density and mechanical strength of the resulting glass-ceramics are listed in Table 4.

## 3. Discussion

The development of glass-ceramic materials based on slag, fly ash, and glass cullet (as the main result of this study) opens many possibilities for future research in this field. Since these glass-ceramic materials are still in the initial phase of development, many of their characteristics are unknown. Future research will have to examine the tensile strength and fracture toughness of the new glass-ceramics. A careful evaluation of the mechanical properties will determine the opportunities for application in construction industries, piezoelectric materials, and others. In this study, during the sintering methods no nucleating agents were used to obtain the glass-ceramics, but future research could consider glass-ceramic materials when using a nucleating agent such as TiO_2_, which could lead to glass-ceramics exhibiting greater mechanical strength.

According SEM results (Figure 1), differences in shape are attributed to the extraction and preparation processes of the materials [23,24]. The elemental composition for each type of fly ash was analyzed using XRF. The main mineralogical component of the fly ash was an aluminosilicate phase that enclosed Fe_2_O_3_, CaO, MgO, and other oxides. Fly ash density normally varies between 1.6 and 3.2 g cm^−3^, according to Ghosal et al. [25].

The slag exhibited the presence of particles that behaved as agglomerates showing similar morphologies (Figure 2a,b) [26]. The particles had a typical angular shape. According to Puertas et al. [27], a normal granulated slag has a vitreous content of 85%–95% by weight, which is favorable for the production of vitreous materials. On the other hand, the glass cullet (Figure 2c) showed the presence of angular particles, exhibiting high heterogeneity in the particle size distribution [23,28]. In conclusion, the raw materials exhibited a vitreous composition also containing crystalline phases being thus of great interest for use in the production of glass-ceramics [26].

The XRF results of the slag samples indicated that approximately 90% of the composition corresponded to Fe, Ca, Si, and Al oxides, with some traces of transition elements, as shown in Figure 4a,b. The slag also exhibited a relatively high content of iron, and the composition was adequate for the production of glass-ceramics, according to previous works [29,30]. In the glass cullet, 90% of the contents corresponded to Si, Ca, and Na oxides, as shown in Figure 4c.

The high silica content in these samples contributed to relatively high viscosity in the material, which could allow for the formation of glassware and glass-ceramics, even if the other components (alkaline oxides) generate a decrease in viscosity. This effect can be compensated for by the aluminum oxide present that acts as network modifier also affecting viscosity [31].

The elemental composition of the three raw materials was evaluated by means of X-ray fluorescence. The fly ash demonstrated the presence of Al, Si, and Fe oxides, identified by their corresponding Kα, confirming that the fly ash samples corresponded to the F classification, according to the ASTM C618-85 (5) standard. The sample showed a content of more than 90% (in weight) of SiO_2_, Al_2_O_3_, and Fe_2_O_3_, as indicated in Figure 3. The presence of titanium oxide was also detected, which could represent an important factor in the formation of glass-ceramics (as a nucleating agent) [8,32,33,34,35,36]. Similarly, the calcium content of around 1.2% and 1.6% (by weight) confirmed the feasibility of promoting devitrification processes and increasing the chemical stability and mechanical behavior of the glasses [6,13].

Taking into account the elemental composition of the industrial wastes studied, a random selection was made for the case of the fly ash (based on the eight types studied), which showed low variability in its composition. The H sample was selected. For the case of slag, the slag 2 sample was selected, which had a higher content of Si, Ca, and Al with respect to slag 1. Finally, the selected glass cullet exhibited a content of Si, Ca, and Na oxides of around 90% of its total composition, as mentioned above.

According to Höland et al. [18], silicates in glass-ceramics can be present in the form of α-quartz, with the following network parameters: a of 4.912(4), c of 5.403(7). The previously identified parameters in this research are important, since they represent a dense polymorph of stable silica at room temperature, which will be useful for obtaining a high hardness in glass-ceramic materials. The content of mullite in the material is important because it is particularly suitable as a refractory material due to its high melting point and its excellent chemical durability. However, if crystallization of the material is not controlled, large undesirable needles of mullite may appear in the glass-ceramics as a result of secondary reactions, which would adversely affect the strength of the material.

According to TGA-DSC analysis (Figure 8, Figure 9 and Figure 10), we can observe that Mixture 1 had a small endothermic signal at 930 °C and an exothermic signal at 1100 °C; and Mixture 2 showed an endothermic signal around 580 °C, which was likely related to a nucleation process, together with an exothermic peak at 800 °C. In the Mixture 3, it is possible to observe a slight endothermic signal at 850 °C and an exothermic peak at 1100 °C, as well as weak thermal events (observed in the DTA analysis) up to 1200 °C, in accordance with other research [37,38,39]. The endothermic signal before the exothermic signal was due to heat absorption accompanied by the softening and rearrangement of the microstructure of the samples during the thermal treatment.

In the TGA curves, no significant weight loss was observed, and the results indicated that the mass loss was less than 0.05 mg for each mixture in the temperature range of 100–1200 °C, indicating that the mixtures had thermal stability in this temperature range [40].

The three mixtures analyzed by XRD presented different crystallographic features (Figure 11, Figure 12 and Figure 13), indicating that the three mixtures had the same crystalline phases, but with different relative contents. In the three glass-ceramic designs, the presence of diopside and anorthite phases in different proportions was evident, according to the Rietveld refinements. The identification of a diopside phase in glass-ceramic materials is important, as this phase contributes to the hardness of the material, in addition to imparting greater chemical resistance to corrosion [26,41]. On the other hand, an anorthite phase has great potential in industrial and technological applications: This crystalline phase is attractive due to its low coefficient of thermal expansion, low dielectric constant, and good wear resistance [42].

Additionally, the presence of an enstatite phase was observed in the three mixtures in different percentages. This phase is commonly used for the preparation of glass-ceramic substrates due to its high strength, low thermal expansion coefficient, and vibration resistance [42]. Using one of the most common compositions in silicate glass-ceramics (composition 50%–85% enstatite), refractory glass-ceramics were produced. This phase is interesting because it represents a martensitic transformation, generating toughening from fracture energy absorption through fine, lamellar twinning [43].

A gehlenite phase appeared in the sintered glass-ceramics in the Mixture 3 that contained 55% slag. The formation of gehlenite was likely the result of a solid reaction of the three wastes used [36,44]. A gehlenite phase is considered beneficial due to the mechanical resistance it can confer to glass-ceramics, which was evaluated for each material. This phase (gehlenite) is a common component of ceramic materials used in construction and for kitchen utensils [45]. In general, four crystalline phases, diopside, anorthite, enstatite, and gehlenite, have been identified in different studies related to the preparation of glass-ceramic materials from industrial waste raw materials: These have been considered for applications in technology, electronics, and construction [46,47,48].

The Mixture 1 showed no changes in its microstructure: That is, no sintering occurred in the material. Meanwhile, in the SEM images of Mixtures 2 and 3, a microstructural change was observed when the material was subjected to the sintering process. The micrographs indicated that the formation of porous aggregates could be related to the release of gases through the elimination of organic substances resulting from the thermal treatment to which the materials were submitted [36].

In the SEM images (Figure 14), pores are observed that could have been associated with spherical particles that induced voids in the matrix when subjected to high temperatures and through the pozzolanic activity of the fly ash. Additionally, the porosity could also have been related to the presence of the enstatite phase in the sintered materials: This transformation usually occurs during the cooling of the material [18], which explains the presence of porosity in samples sintered at high temperatures (the relatively high porosity will lead to low mechanical properties) [49,50].

The Mixture 2 had a higher fracture strength (10.8 MPa), which could be attributed to the crystalline phases identified (diopside 73% and anorthite 27%). The phase of enstatite was identified in Mixtures 1 and 3, possibly leading to lower strength with respect to sample 2, where this phase was not present [18,26,42]. The average fracture strength values measured on the current glass-ceramics were 3 MPa for Mixture 1, 10.8 MPa for Mixture 2, and 4 MPa for Mixture 3. The fracture strength of Mixture 2 was (approximately) more than triple that of Mixture 1 and more than double that of Mixture 3, which had a fracture strength similar to that of glass-ceramics in the CaO–Al_2_O_3_–SiO_2_ system reported in the literature [51] (Table 4). Additionally, the ASTM C28 (2015) norm indicates that values in the range 3.1–8.3 MPa are relevant for applications of gypsum materials in construction [50] and concrete. Since differences in the crystalline phases in Mixtures 1, 2, and 3 exist, and the types associated with diopside in Mixtures 2 and 3 and anorthite in Mixture 1, the difference in mechanical strength values could be attributed to the different main crystalline phases present in the different samples. It has been reported that diopside is a more preferable crystalline phase than anorthite from the point of view of the mechanical properties of glass-ceramics [52]. It has also been suggested that diopside is a preferable crystalline phase in glass-ceramics compared to wollastonite in terms of mechanical properties [51].

The apparent density was determined through the Archimedes method (immersion in water). Table 4 shows the results obtained, which were comparable to previous investigations where glass-ceramic materials were obtained from industrial waste (1.6–3.2 g/cm^−3^) [8,25,36,41,53,54]. In samples 1 and 2, the density was lower, which could have been related to the percentage by weight of glass in the starting mixtures.

## 4. Materials and Methods

The current work focused on an analysis of eight types of fly ashes labeled A, B, C, D, E, F, G, or M; two types of slags (slag 1, slag 2), and one sample of glass cullet. These three waste materials were morphologically characterized, and the best candidate for the fabrication of glass-ceramics was selected from each type. According to the characterization results, the selected fly ash was the one identified as M, and it exhibited an elemental composition low in Fe (10.1 wt %) and S (0.77 wt %), high in Al (25.8 wt %) and Si (53.7 wt %), and regular morphology. Even if the fly ash identified as E also exhibited an elemental composition low in Fe and S and high in Al and Si, the microstructure of the E fly ash particles (cenospheres) showed irregular morphology. For this reason, the H fly ash was selected. Regarding the selected slag, the collected sample was evaluated in terms of its chemical composition by X-ray fluorescence (XRF). The results confirmed a low content of Fe and high contents of Si, Ca, Al, and Mg oxides, materials that could form glass and that are necessary to consolidate glass-ceramics [1]. The used glass cullet was characterized by a composition containing Si, Ca, and Na oxides, a composition that has previously demonstrated the feasibility to obtain different silicate materials, including glass-ceramics, with interesting properties [8,26,30,32,33].

The different types of waste materials used in this research were pretreated and cleaned to avoid contamination using a simple process of water washing. After that, the raw materials of slag, fly ash, and glass cullet were separately grounded, and particle sizes below 75 µm for the slag and fly ash and below 149 µm for the glass cullet were obtained. A loss-on-ignition (LOI) process to remove the content of unburned carbon in the fly ash samples was carried out following the procedure described in previous works [23]. Subsequently, the three materials were mixed according to the mixture compositions defined in Table 1. The glass-ceramic materials synthesis was carried out under a controlled Ar atmosphere.

The elemental analysis performed by X-ray fluorescence (XRF) was conducted in PANalitical MiniPal 2 spectrometer equipment operated at 20 KeV. The structural analysis was performed by X-ray diffraction (XRD) on powder samples in a Bruker D8 Advance (Karlsruhe, Germany) diffractometer using CuK_α_ radiation (1.5418 Å) with 2Ɵ in the range 10–50° and a step size of 0.020°. Phase identification was performed by means of X’Pert High Score software in the ICDD databases. Thermogravimetric analysis (DTA/TGA) was performed in a standard SDT Q600 20 DSC-TGA instrument (TA Instruments, New Castle, DE, USA) using a heating rate of 10 °C min^−1^ under argon flow conditions (100 mL min^−1^) from room temperature up to 1100 °C. A qualitative morphological evaluation and surface characterization of materials was carried out using scanning electron microscopy (SEM) equipped with an energy-dispersed X-ray (SEM-EDX) detector (LEO 435 Electron Microscope Ltd., Cambridge, UK, and Ultra Plus, Zeiss, Jena, Germany). The geometric density was determined using the Archimedes method, characterizing the relationship between the mass and volume of samples. The mass was determined using a balance (AEJ 220-M KERN, Balingen, Germany with a resolution of 0.001 g cm^−3^. The compression strength of pellet samples was measured using a universal testing machine (Zwick Roell, series Z050, Ulm, Germany), with a loading speed of 0.5 mm min^−1^, and a minimum of 5 samples for each composition were measured.

## 5. Conclusions

A new family of glass-ceramic materials was obtained from combinations of three types of silicate waste. The process to obtain glass-ceramic materials from mixtures of wastes was investigated by considering DTA results to determine the right heating treatment to achieve high crystallinity. In this research, materials with crystalline phases (diopside, anorthite, enstatite, and gehlenite) were obtained. The results derived from fracture strength experiments by the Brazilian strength test allowed us to determine values below 10.8 MPa, which represents a limitation for structural applications. However, the results were similar to previous research on glass-ceramics obtained from industrial waste, and moreover, the density calculations for each sample were found to be in agreement with previous investigations (1.6–3.2 cm^−3^) involving different raw materials (slag, glass cullet, fly ash). The present results confirm the feasibility of the reuse of industrial waste from the Boyacá, Colombia, region to obtain sound glass-ceramic materials for potential applications in construction, which must be however further optimized to achieve higher mechanical strength.

## Figures and Tables

**Figure 1 materials-12-02032-f001:**
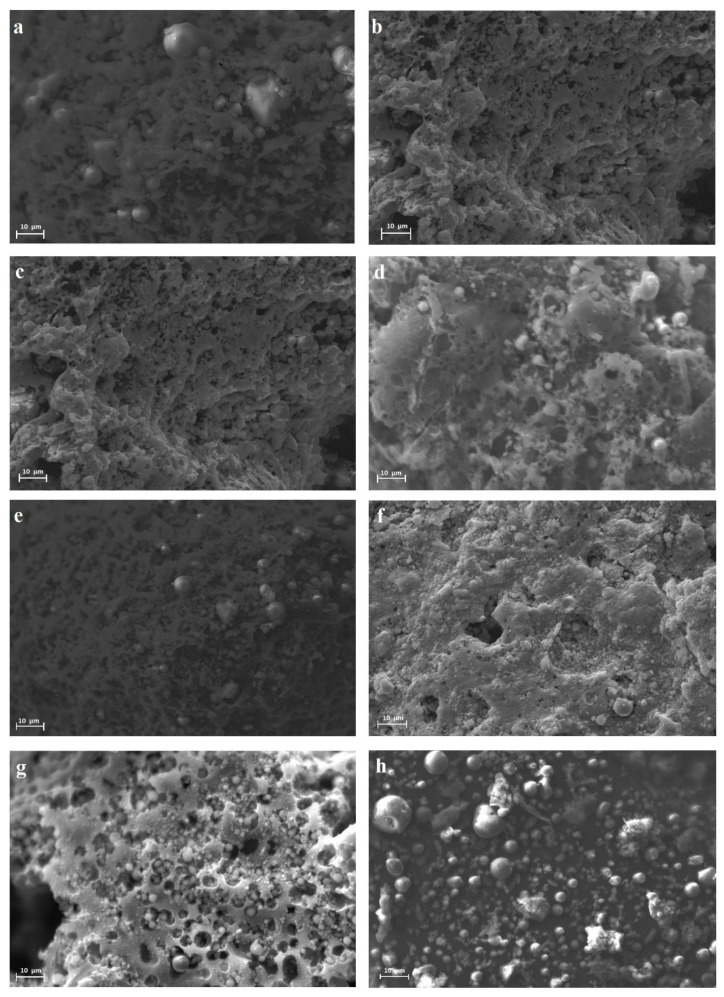
Scanning electron microscopy (SEM) images for: (**a**) fly ash A, (**b**) fly ash B, (**c**) fly ash C, (**d**) fly ash D, (**e**) fly ash E, (**f**) fly ash F, (**g**) fly ash G, and (**h**) fly ash H.

**Figure 2 materials-12-02032-f002:**
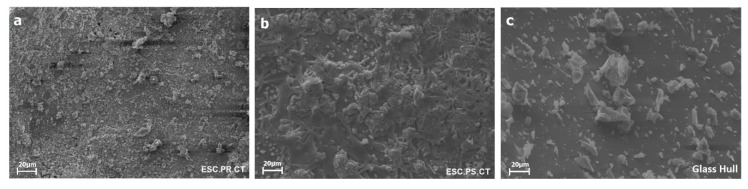
SEM images for (**a**) slag 1, (**b**) slag 2, and (**c**) glass cullet.

**Figure 3 materials-12-02032-f003:**
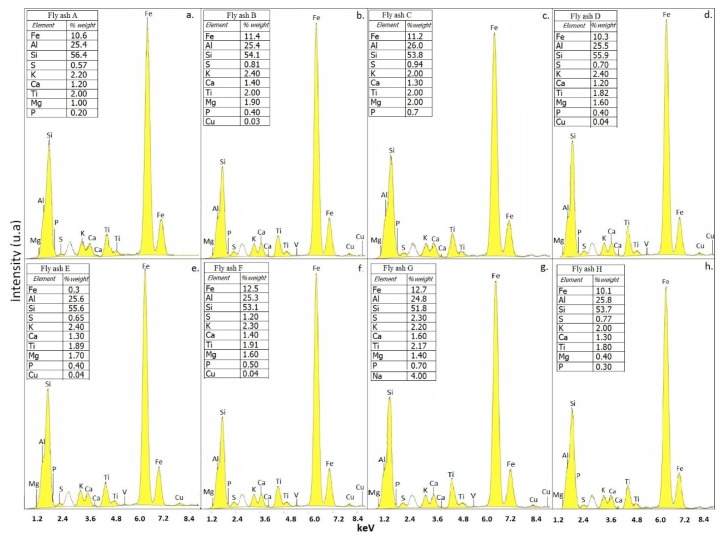
X-ray fluorescence analysis of precursor materials associated with (**a**) fly ash A, (**b**) fly ash B, (**c**) fly ash C, (**d**) fly ash D, (**e**) fly ash E, (**f**) fly ash F, (**g**) fly ash G, and (**h**) fly ash H.

**Figure 4 materials-12-02032-f004:**
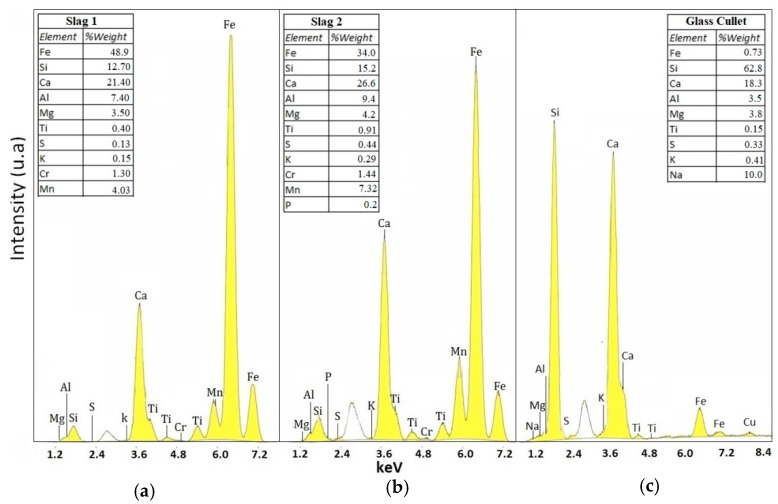
X-ray fluorescence analysis of elemental composition of precursor materials associated with (**a**) slag 1, (**b**) slag 2, and (**c**) glass cullet.

**Figure 5 materials-12-02032-f005:**
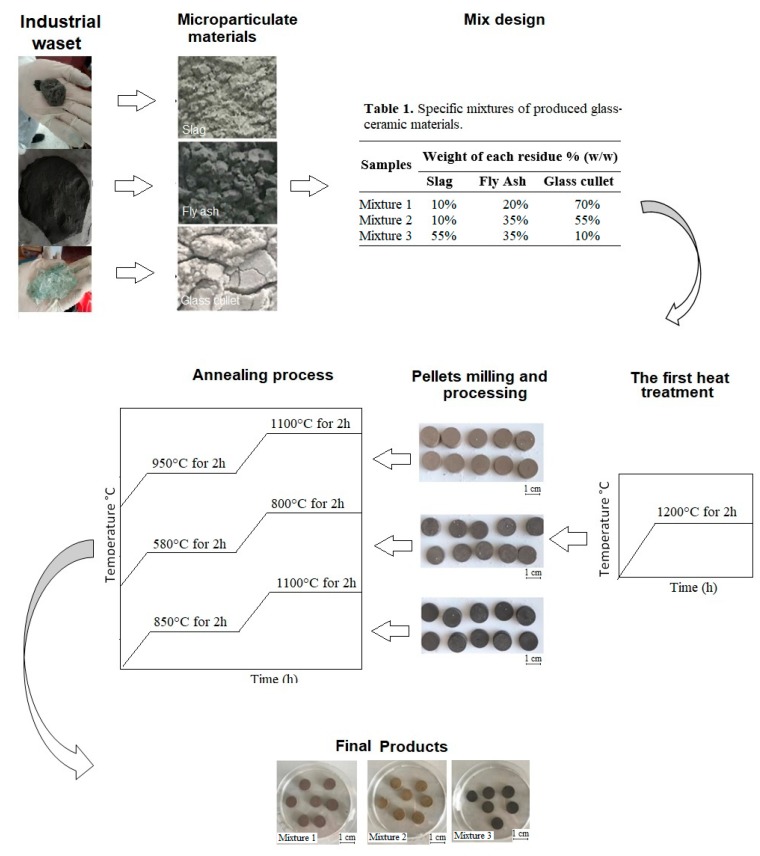
Schematic diagram showing the heat treatment and annealing processes for the production of glass-ceramics: Mixture 1, Mixture 2, and Mixture 3.

**Figure 6 materials-12-02032-f006:**
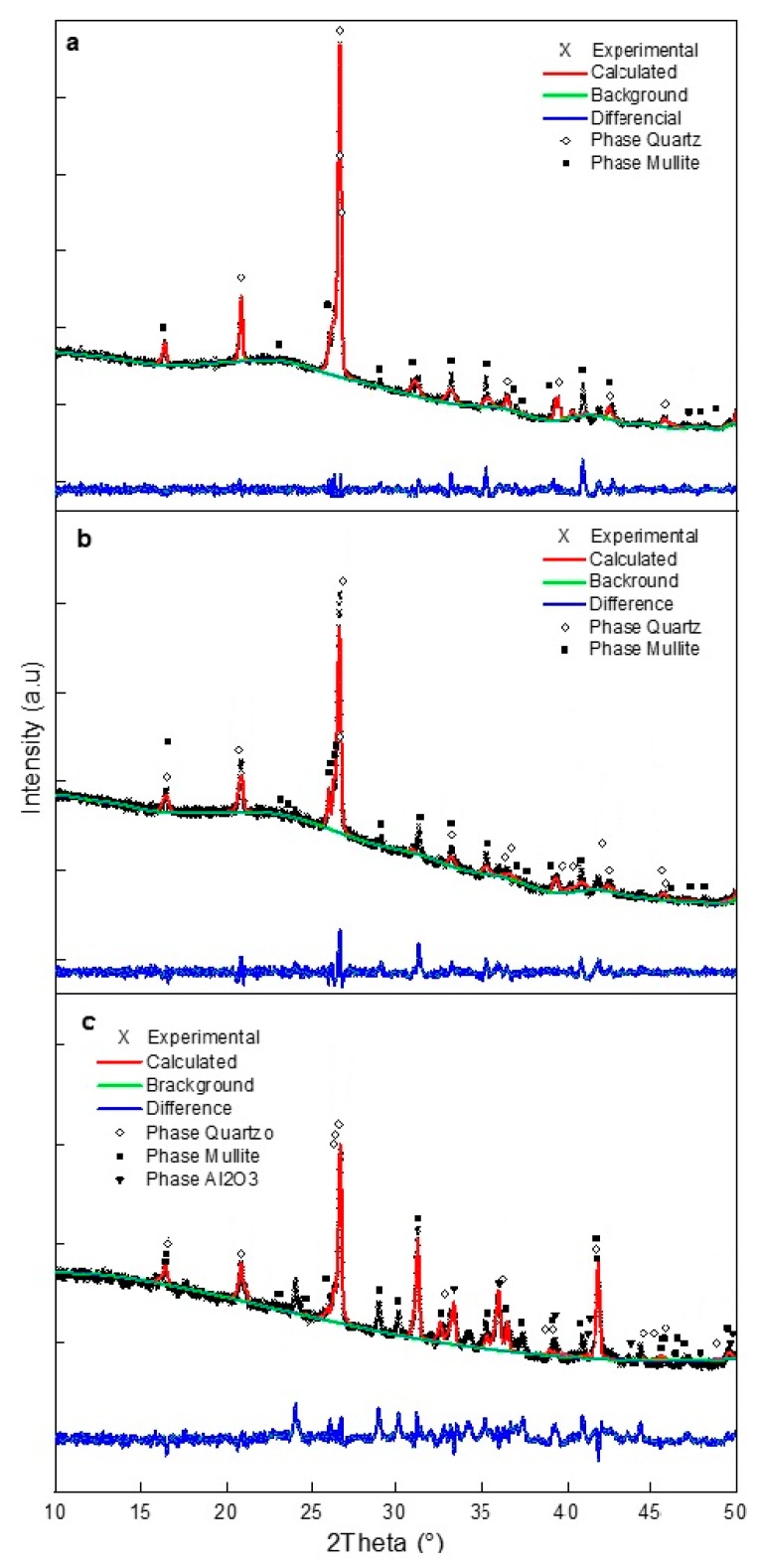
X-ray diffraction patterns of (**a**) Mixture 1, (**b**) Mixture 2, and (**c**) Mixture 3 with corresponding Rietveld refinement.

**Figure 7 materials-12-02032-f007:**
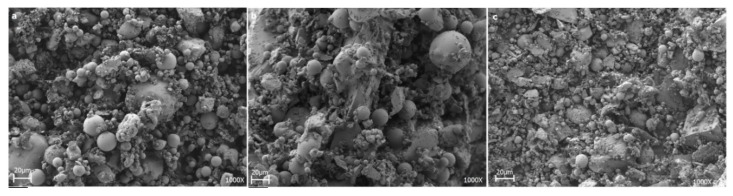
Scanning electron microscopy images: (**a**) Mixture 1, (**b**) Mixture 2, and (**c**) Mixture 3 prior to thermal treatments.

**Figure 8 materials-12-02032-f008:**
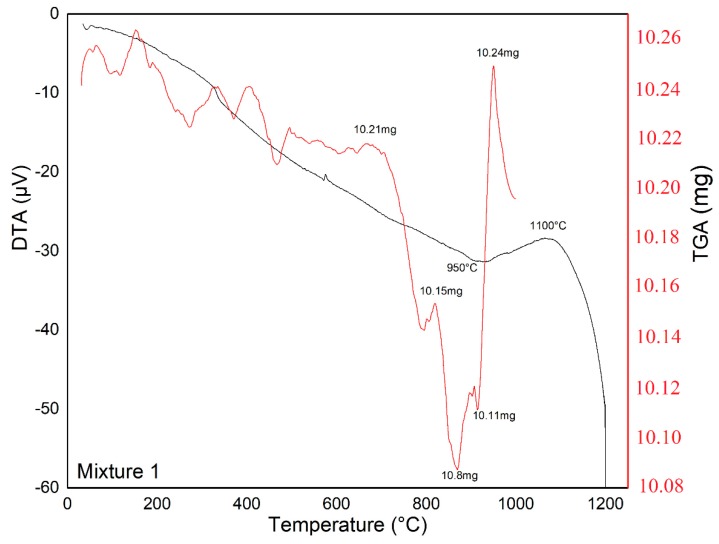
Thermal analysis (DTA-TGA) between 25 and 1200 °C for Mixture 1 showing characteristic temperatures.

**Figure 9 materials-12-02032-f009:**
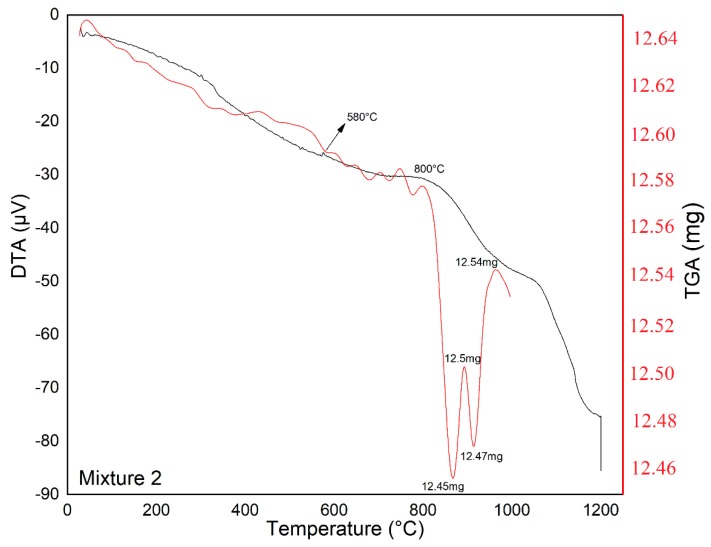
Thermal analysis (DTA-TGA) between 25 and 1200 °C for Mixture 2 showing characteristic temperatures.

**Figure 10 materials-12-02032-f010:**
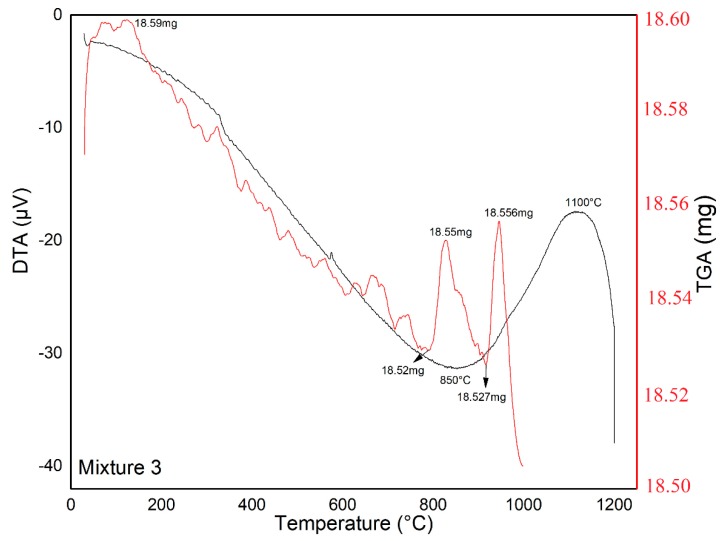
Thermal analysis (DTA-TGA) between 25 and 1200 °C for Mixture 3 showing characteristic temperatures.

**Figure 11 materials-12-02032-f011:**
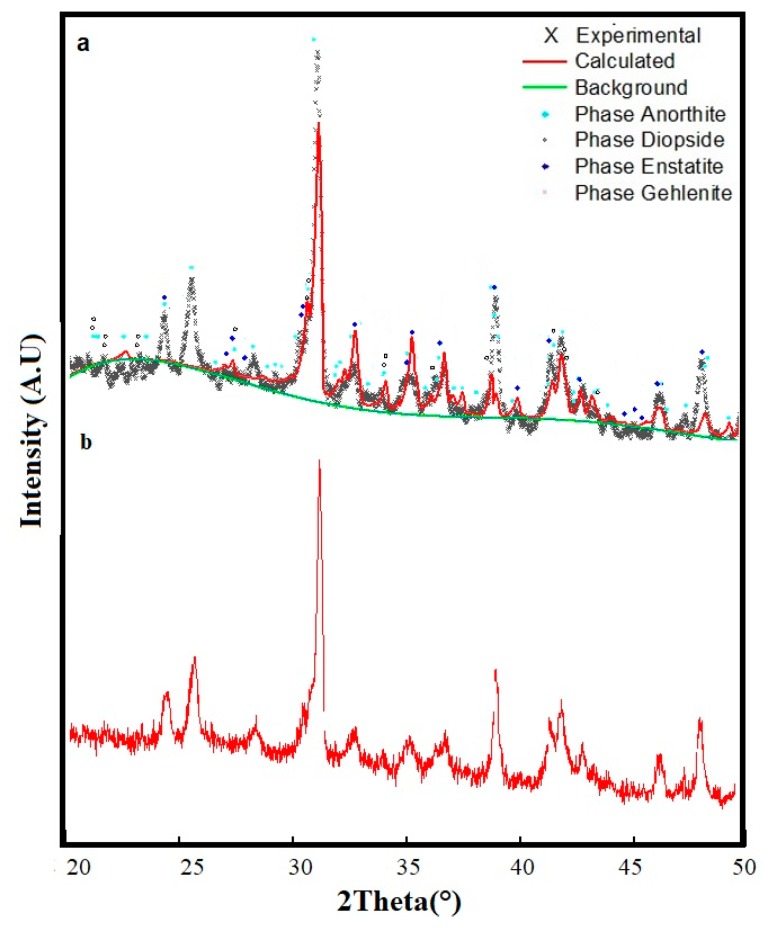
X-ray diffraction patterns: (**a**) Mixture 1 at 1200 °C with the annealing process at 930 °C and 1100 °C for 2 h; (**b**) Mixture 1 at 1200 °C.

**Figure 12 materials-12-02032-f012:**
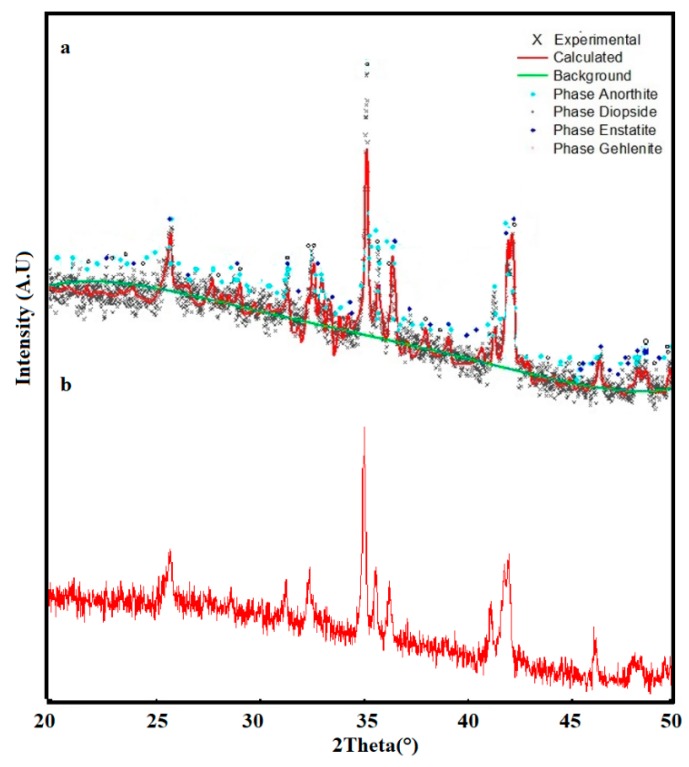
X-ray diffraction patterns: (**a**) Mixture 2 at 1200 °C with the annealing process at 930 °C and 1100 °C for 2 h; (**b**) Mixture 2 at 1200 °C.

**Figure 13 materials-12-02032-f013:**
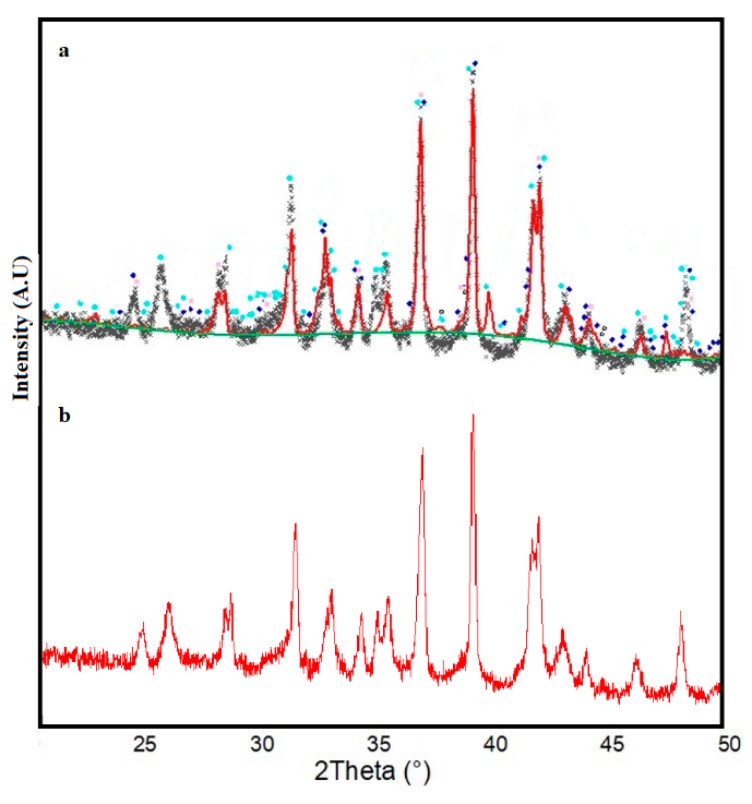
X-ray diffraction patterns: (**a**) Mixture 3 at 1200 °C with the annealing process at 930 °C and 1100 °C for 2 h; (**b**) Mixture 3 at only 1200 °C.

**Figure 14 materials-12-02032-f014:**
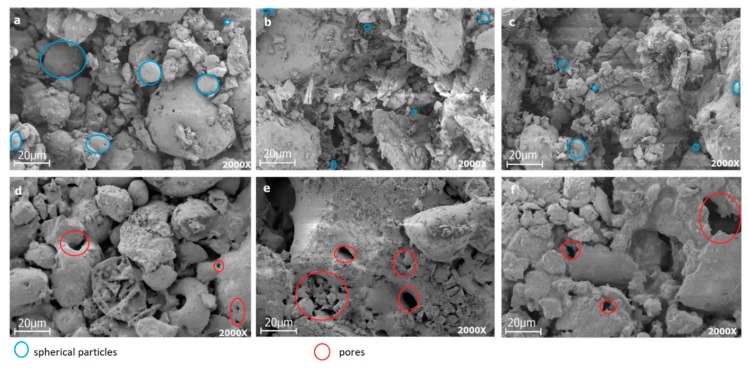
Scanning electron microscopy images at 2000×: (**a**) Mixture 1 at 1200 °C, (**b**) mixture 2 at 1200 °C, (**c**) mixture 3 at 1200 °C, (**d**) mixture 1 at 1200 °C with the annealing process at 930 °C and 1100 °C for 2 h, (**e**) mixture 2 at 1200 °C with the annealing process at 930 °C and 1100 °C for 2 h, (**f**) mixture 3 at 1200 °C with the annealing process at 930 °C and 1100 °C for 2 h.

**Table 1 materials-12-02032-t001:** Specific mixtures of the produced glass-ceramic materials.

Samples	Weight of Each Residue % (w/w)		
	Slag	Fly Ash	Glass Cullet
Mixture 1	10%	20%	70%
Mixture 2	10%	35%	55%
Mixture 3	55%	35%	10%

**Table 2 materials-12-02032-t002:** Composition of mixtures investigated (mol percent).

Samples	Na_2_O	MgO	Al_2_O_3_	SiO_2_	SO_3_	K_2_O	CaO	TiO_2_	Fe_2_O_3_	Mn	Other
Mixture 1	3.627	1.85	13.64	66.3	0.8	0.7	7.1	1.09	3.924	0.7	0.285
Mixture 2	4.914	3.38	8.161	66.4	0.6	0.4	12	0.64	2.828	0.7	0.227
Mixture 3	1.813	3.12	11.24	46.9	0.7	0.5	20	1.03	9.686	3.7	0.927

**Table 3 materials-12-02032-t003:** Content temperatures for sintering treatments according to the DTA results.

Samples	Melting Temperature (°C)	Nucleation Temperature (°C)	Crystallization Temperature (°C)	Time (h)
Mixture 1	1200	930	1100	2
Mixture 2	1200	580	800	2
Mixture 3	1200	850	1100	2

**Table 4 materials-12-02032-t004:** Results of mechanical property determinations.

Sample	Density (g cm^−3^)	Strength (MPa)	According to Literature (MPa)
Mixture 1	2.26	3 ± 1	0.4–6.0 [21] 1.77–12.21 [22]
Mixture 2	2.43	10.8 ± 0.4	
Mixture 3	2.73	4 ± 2	

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
