# Peer review of "Development and Characterization of Glass-Ceramics from Combinations of Slag, Fly Ash, and Glass Cullet without Adding Nucleating Agents"

_materials, 2019, doi:10.3390/ma12122032_

Reviewer 1 Report

The manuscript reports on  the preparation and (SEM, TGA-DTA, XRD, Brazilian compressive test) characterization of glass-ceramic pellets from slag, fly ash and glass cullet collected in Colombia.

The recycling of industrial waste is an important issue studied since decades (as demonstrated by the 2006 review paper of one of the authors, ref 8) and new way to use these waste is interesting. The paper may be published after the following weaknesses have been addressed.

The novelty of the work must be clearly explained in the text, in the abstract and if possible in the title. Quantitative information must be given in the abstract (typical composition, melting  and sintering temperature, mechanical strength, …).

Materials and Methods: information must be given in this paragraph and not distributed in Results paragraph.

Quality of many figures is poor: increase the contrast of images in Figs 1 and 2; use darker XRF spectrum lines

A table comparing the elemental composition of A to M fly ash, slag and cullet will be more readable than Figs 3 and 4 small inserts.

Layouts of Figs 6 and 11 are strange; why bend peaks; show clearly the experimental data and compare with reference data as usual; note some peaks are not identified, as exampled for Fig. 6c, 11a; the correspondence between experimental data and reference phases is not good.

Table 3: add melting temperature.

Table 4: I don’t see interest of pellet diameter and thickness in the Table (to be given in Experimental part).

Merge Discussion and conclusion in a single paragraph. And delete the lines 314-315!!!!

Author Response

Response to Reviewer 1 Comments

Point 1: The novelty of the work must be clearly explained in the text, in the abstract and if possible in the title. Quantitative information must be given in the abstract (typical composition, melting and sintering temperature, mechanical strength, …).

Response 1: The innovation part was explained in the abstract, it has the composition, melting temperature and sintering in the abstract.

Point 2: Materials and Methods: information must be given in this paragraph and not distributed in Results paragraph.

Response 2: The title "Experimental process" was changed by "materials and methods" and the titles of the different techniques indicated in the text were eliminated from all content for greater understanding.

Point 3: Quality of many figures is poor: increase the contrast of images in Figs 1 and 2; use darker XRF spectrum lines.

Response 3: The figures were adjusted

Point 4: A table comparing the elemental composition of A to M fly ash, slag and cullet will be more readable than Figs 3 and 4 small inserts.

Response 4: The table in the figures were adjusted

Point 5: Layouts of Figs 6 and 11 are strange; why bend peaks; show clearly the experimental data and compare with reference data as usual; note some peaks are not identified, as exampled for Fig. 6c, 11a; the correspondence between experimental data and reference phases is not good.

Response 5: The figures were adjusted

Point 6: Table 3: add melting temperature.

Response 6: we add the melting temperature in table 3

Point 7: Table 4: I don’t see interest of pellet diameter and thickness in the Table (to be given in Experimental part).

Response 7: we delete the diameter and thickness in the table.

Point 8: Merge Discussion and conclusion in a single paragraph. And delete the lines 314-315!!!

Response 8: we do not make changes because in the template of the magazine, they ask to have the two aspects separately.

Reviewer 2 Report

The paper presents the results of synthesis of glass-ceramics using slag, fly ash and glass cullet as raw materials aiming to achieve glass-ceramic products with attractive technical properties for potential industrial applications. The manuscript presents all important data on raw materials analysis, synthesis of glass-ceramics and characterisation of materials obtained. The paper is of good quality and is practically ready for publication with just two comments of attention for authors:

(1)   Authors concluded that the glass-ceramics obtained are characterised by “mechanical properties for applications in the construction industry” although the data from Table 4 would need a comparison with construction standard requirements or construction materials conventionally used for the confidence of conclusion.

(2)   Additionally, in the introductory part of paper it is worth noting that glass-ceramics are versatile wasteforms for immobilising various types of both radioactive and hazardous wastes that was demonstrated in many works including that co-authored by one of paper authors. Compared to homogeneous glassy materials glass-ceramics can incorporate larger amounts of waste elements and are produced using lower processing temperatures which is particularly important for nuclear waste immobilisation technologies [M.I. Ojovan, J.M. Juoi, A.R. Boccaccini, W.E. Lee. Glass Composite Materials for Nuclear and Hazardous Waste Immobilisation. Mater. Res. Soc. Symp. Proc., 1107, 245-252 (2008)].  

Author Response

Response to Reviewer 2 Comments

Point 1:  Authors concluded that the glass-ceramics obtained are characterised by “mechanical properties for applications in the construction industry” although the data from Table 4 would need a comparison with construction standard requirements or construction materials conventionally used for the confidence of conclusion.

Response 1: The results were correlated with other investigations where they obtained glass-ceramic materials with these properties, however, also correlated whit ASTM C28 (2015) was added.

Point 2: Additionally, in the introductory part of paper it is worth noting that glass-ceramics are versatile wasteforms for immobilising various types of both radioactive and hazardous wastes that was demonstrated in many works including that co-authored by one of paper authors. Compared to homogeneous glassy materials glass-ceramics can incorporate larger amounts of waste elements and are produced using lower processing temperatures which is particularly important for nuclear waste immobilisation technologies [M.I. Ojovan, J.M. Juoi, A.R. Boccaccini, W.E. Lee. Glass Composite Materials for Nuclear and Hazardous Waste Immobilisation. Mater. Res. Soc. Symp. Proc., 1107, 245-252 (2008)].

Response 2: the reference suggested by the jury was included

Reviewer 3 Report

The paper is a highly valid study on the use of glass cullet in recycled products.  The paper is very thorough and is of sound scientific value.  However, the authors should conduct a thorough spelling and grammar check prior to publication.

Author Response

Point 1: The paper is a highly valid study on the use of glass cullet in recycled products.  The paper is very thorough and is of sound scientific value.  However, the authors should conduct a thorough spelling and grammar check prior to publication.

Response 1: Thanks, we did make in the document a revision and correction of spelling and grammar.

Reviewer 4 Report

The manuscript describes development and characterization of glass-ceramics from combinations of slag, fly ash and glass cullet.

The manuscript requires major revision. Its structure should be reorganized because “Materials and Methods” section should go after the “Discussion” one (please see Materials — Instructions for Authors, https://www.mdpi.com/journal/materials/instructions#preparation). The Discussion section is surprisingly short (three lines). In instructions for Authors, it is written, “Authors should discuss the results and how they can be interpreted in perspective of previous studies and of the working hypotheses. The findings and their implications should be discussed in the broadest context possible and limitations of the work highlighted. Future research directions may also be mentioned.”

P. 15, line 314. Please explain the reason for the appearance of the sentence “This section is not mandatory, but can be added to the manuscript if the discussion is unusually long or complex.”

P. 2, lines 56-69. This paragraph should be placed to Materials and methods section.

Please present XRD patterns of mixtures after their first heat-treatment.

Technical comments are listed below.

P. 1. The abbreviation XRF is used on p. 1 for the first time. It is explained only on p. 3. Please make correction.

P. 3, line 116. Please correct typo in the phrase “The morphology the of raw materials…”

P. 7, line 177. Please correct typo in the phrase “In these Figureures,”

Author Response

Response to Reviewer 4 Comments

Point 1: The manuscript requires major revision. Its structure should be reorganized because “Materials and Methods” section should go after the “Discussion” one (please see Materials — Instructions for Authors, https://www.mdpi.com/journal/materials/instructions#preparation). The Discussion section is surprisingly short (three lines). In instructions for Authors, it is written, “Authors should discuss the results and how they can be interpreted in perspective of previous studies and of the working hypotheses. The findings and their implications should be discussed in the broadest context possible and limitations of the work highlighted. Future research directions may also be mentioned.”

Response 1: the discussion information was supplemented

Point 2: P. 15, line 314. Please explain the reason for the appearance of the sentence “This section is not mandatory, but can be added to the manuscript if the discussion is unusually long or complex.”

Response 2: This sentence has been deleted

Point 3: P. 2, lines 56-69. This paragraph should be placed to Materials and methods section.

Response 3: we made the suggested adjustment.

Point 4: Please present XRD patterns of mixtures after their first heat-treatment.

Response 4: In figures 11 12 and 13 the XRD patterns of the mixtures were added with the first heat treatment

Point 5: P. 1. The abbreviation XRF is used on p. 1 for the first time. It is explained only on p. 3. Please make correction.

Response 5: On page 7 the XRF technique was explained, on page 8 and 9 the analysis results are explained.

Point 6: P. 3, line 116. Please correct typo in the phrase “The morphology the of raw materials…”

Response 6: we made the suggested adjustment

Point 7: P. 7, line 177. Please correct typo in the phrase “In these Figureures,”

Response 7: we made the suggested adjustment

Round  2

Reviewer 1 Report

The authors has improved/clarified the text. It deserves publication after the quality of Fig. 3 has been improved (increase the contrast )

Author Response

Response to Reviewer 1 Comments

Point 1: The authors has improved/clarified the text. It deserves publication after the quality of Fig. 3 has been improved (increase the contrast)

Response 1: We changed figure 3

Reviewer 4 Report

The following sentences require grammar correction:

P. 3, line 65. And the morphology of the  slag and glass cullet, can you observed in the Fig 2.

P. 4, line 93. The elemental composition of the three raw materials was evaluated by means of X-ray  fluorescence (Fig 3) for the case of the slag and glass cullet is shown in Fig 4.

P. 10, line 166. Figure 11. X-ray diffraction patterns of the mixture 1 a. X1-1200°C whit the annealing  process, b. only to 1200°C.

P. 14, line 251. The corresponding TGA-DSC analyses (Figs 8-10), is clear that X1 mixture has a small endothermic signal at 930°C and an exothermic signal at 1100°C, the X2 mixture exhibits one endothermic signal around 580°C, which is related to a nucleation process, together with the  exothermic peak at 800°C.

P. 14, line 254. In accordance with Harabi et al.[37], in the X3 mixture, it is  possible to observe a slight endothermic signal at 850°C and an exothermic peak at 1100°C, as well on weak thermal events observed in DTA analysis up to 1200°C [38,39].

Author Response

Response to Reviewer 4 Comments

Point 1: P. 3, line 65. And the morphology of the slag and glass cullet, can you observed in the Fig 2.

Response 1: the grammatical correction was made

Point 2: P. 4, line 93. The elemental composition of the three raw materials was evaluated by means of X-ray fluorescence (Fig 3) for the case of the slag and glass cullet is shown in Fig 4.

Response 2: the grammatical correction was made

Point 3: P. 10, line 166. Figure 11. X-ray diffraction patterns of the mixture 1 a. X1-1200°C whit the annealing  process, b. only to 1200°C.

Response 3: the grammatical correction was made

Point 4: P. 14, line 251. The corresponding TGA-DSC analyses (Figs 8-10), is clear that X1 mixture has a small endothermic signal at 930°C and an exothermic signal at 1100°C, the X2 mixture exhibits one endothermic signal around 580°C, which is related to a nucleation process, together with the exothermic peak at 800°C

Response 4: the grammatical correction was made

Point 5: P. 14, line 254. In accordance with Harabi et al.[37], in the X3 mixture, it is  possible to observe a slight endothermic signal at 850°C and an exothermic peak at 1100°C, as well on weak thermal events observed in DTA analysis up to 1200°C [38,39].

Response 5: the grammatical correction was made